# Comparative Analysis of Minimum Chip Thickness, Surface Quality and Burr Formation in Micro-Milling of Wrought and Selective Laser Melted Ti64

**DOI:** 10.3390/mi14061160

**Published:** 2023-05-30

**Authors:** Uçan Karakılınç, Berkay Ergene, Bekir Yalçın, Kubilay Aslantaş, Ali Erçetin

**Affiliations:** 1Department of Computer Programming, Isparta University of Applied Sciences, 32200 Isparta, Turkey; ucankarakilinc@isparta.edu.tr; 2Department of Mechanical Engineering, Pamukkale University, 20160 Denizli, Turkey; bergene@pau.edu.tr; 3Department of Mechanical Engineering, Afyon Kocatepe University, 03200 Afyonkarahisar, Turkey; aslantas@aku.edu.tr; 4Department of Naval Architecture and Marine Engineering, Faculty of Maritime, Bandırma Onyedi Eylul University, 10200 Bandırma, Turkey; aercetin@bandirma.edu.tr

**Keywords:** selective laser melting, casting, Ti64, micro-milling, surface quality

## Abstract

Selective laser melting (SLM) is a three-dimensional (3D) printing process that can manufacture functional parts with complex geometries as an alternative to using traditional processes, such as machining wrought metal. If precision and a high surface finish are required, particularly for creating miniature channels or geometries smaller than 1 mm, the fabricated parts can be further machined. Therefore, micro milling plays a significant role in the production of such miniscule geometries. This experimental study compares the micro machinability of Ti-6Al-4V (Ti64) parts produced via SLM compared with wrought Ti64. The aim is to investigate the effect of micro milling parameters on the resulting cutting forces (F_x_, F_y_, and F_z_), surface roughness (R_a_ and R_z_), and burr width. In the study, a wide range of feed rates was considered to determine the minimum chip thickness. Additionally, the effects of the depth of cut and spindle speed were observed by taking into account four different parameters. The manufacturing method for the Ti64 alloy does not affect the minimum chip thickness (MCT) and the MCT for both the SLM and wrought is 1 μm/tooth. SLM parts exhibit acicular α martensitic grains, which result in higher hardness and tensile strength. This phenomenon prolongs the transition zone of micro-milling for the formation of minimum chip thickness. Additionally, the average cutting force values for SLM and wrought Ti64 fluctuated between 0.072 N and 1.96 N, depending on the micro milling parameters used. Finally, it is worth noting that micro-milled SLM workpieces exhibit lower areal surface roughness than wrought ones.

## 1. Introduction

Ti-6Al-4V alloy, also known as Ti64, is a type of α + β titanium alloy that finds widespread use in various sectors such as the biomedical, automobile, telecommunications, marine, and defense industries. This is primarily due to its exceptional material properties, such as high specific strength, excellent corrosion resistance, and biocompatibility [1,2,3,4]. However, the production of Ti64 poses challenges due to its low thermal conductivity [5], its tendency to undergo active chemical reactions with oxygen [6], and its susceptibility to strain hardening [7,8]; despite these, there is a high demand for Ti64 products. Conventional methods, such as powder metallurgy [9,10], rolling, casting, and the forging of bulk feedstock materials, can be used to manufacture Ti64 products [11]. However, these traditional methods often lead to high production costs, long lead times, and significant material waste [12].

The Selective Laser Melting (SLM) method belongs to the powder bed fusion (PBF) class, where metallic powder layers are solidified based on pre-generated g-codes using a slicing program until the desired part is fully fabricated [13,14,15]. SLM offers a high degree of flexibility in producing metallic parts with intricate shapes and geometries [16,17]. Examples of such parts include hip, knee, and cranial implants [18] as well as scaffolds [19,20], crash boxes [21], aircraft wing components [22], and energy absorber elements [23]. While SLM allows for the production of unique products with complex geometric shapes, including lattice structures and partially filled cellular structures, additional machining operations are often required to achieve a satisfactory surface finish due to the nature of the laser-based PBF process. For instance, miniature features such as micro channels can be challenging to use in the SLM process, thus necessitating the use of micro-milling processes.

In the literature, numerous studies have focused on the wear or mechanical performance of the Ti64 parts manufactured using SLM [24,25,26,27,28]. For instance, one study conducted an analysis of standard Ti64 and SLM Ti64 in turning operations, examining cutting forces and surface roughness [29]. They observed that cutting forces were higher for SLM Ti64 compared to cast Ti64 due to its increased strength, while SLM Ti64 exhibited lower surface roughness due to its increased brittleness and hardness. The micro-milling of additive manufacturing-produced Ti64 parts was also investigated, analyzing burr formation, surface roughness, and cutting forces [30]. It was found that AM Ti64, with its finer microstructure, displayed higher hardness and lower cutting forces compared to as-cast Ti64. Hojati et al. [31] highlighted the differences in machining Ti alloys produced via electron beam melting (EBM), SLM, and as-cast processes by evaluating the surface roughness, cutting forces, and burr formation. Their results showed that EBM titanium exhibited higher hardness with similar or lower cutting forces compared to conventional titanium. Coz et al. [32] conducted a comparison of the machinability of SLM-produced Ti64 and casting processes. During the micro-cutting tests, the influence of the operation conditions on chip formation, cutting forces, and the surface and sub-surface microstructure were analyzed. Their findings revealed that the parts exhibited higher cutting forces (from 3% to 24% more). Khaliq et al. [33] investigated the tool wear, surface quality, and residual stress during the micro-milling of 3D-printed Ti64 using SLM. They observed that, at lower feed rates where elastoplastic deformation occurs, a plowing effect induced increased residual stresses. Campos et al. [34] conducted a comparative investigation on the micro-milling of Ti64 manufactured by the conventional method and SLM. They found that the SLM parts exhibited an α martensite structure with finer acicular grains, and higher strength and hardness compared to the wrought samples. Despite the finer microstructure and higher hardness, SLM titanium alloy exhibited a 9.3% lower cutting force compared to wrought titanium alloy [35].

In this study, the machinability of Ti64 parts produced through casting and the SLM method were compared using micromachining processes, focusing on cutting forces, surface roughness, and burr formation. The minimum chip thickness (MCT) was determined for both materials using a wide range of cutting parameters. The study considered not only variations in cutting forces but also variations in top burr width and surface roughness when determining the MCT. Through a detailed experimental study, the study aimed to reveal the effects of the production method of the Ti64 alloy on the MCT.

## 2. Characterization of Ti64 Materials

In this study, two different materials were utilized: Ti64 alloy produced by SLM and wrought Ti64 alloy. The wrought Ti64 was supplied by a company, and the SLM Ti64 workpiece was produced using the laser sintering system EOSINT M280 SLM machine. The EOSINT M280 machine has a Yb fiber laser that can operate at up to 200 W, with the highest scanning speed being 7 m/s, and offers a variable laser diameter between 100 μm and 500 μm. For the production of the Ti64 part with SLM, Ti64 spherical powders with diameters ranging between 30 and 50 μm are preferred. The chemical and physical properties of the powders are given in Table 1. Additionally, during the production steps of the workpiece, the advised process parameters provided by the manufacturing company were used (Table 2).

Moreover, a stress relief process was applied to SLM Ti64 parts, which involved subjecting them to 650 °C for 3 h. This process aimed to minimize residual stresses and prevent warpage resulting from the repetitive heating and cooling cycles during the production stage. Lastly, Figure 1, below, illustrates the Ti64 powder’s particles.

### 2.1. Microstructure Analysis of the Fabricated Ti64 Parts

The microstructural analyses of the produced Ti64 parts were conducted using the FEI Quanta Feg 250 model scanning electron microscope (SEM). The metallographic samples were polished using polishing papers with 400, 1000, and 1500 mesh sizes, respectively. The polishing procedure was completed using diamond paste. Subsequently, an etching solution consisting of 85% H_2_O, 10% HF, and 5% HNO_3_ was prepared, and etching was performed for 5 min. The microstructure and energy dispersive spectrometry (EDS) of the wrought Ti64 part are shown in Figure 2a. The obtained microstructure revealed the presence of α and α-β grains, which aligns with observations from another study [36]. Additionally, Figure 2b displays the presence of acicular α martensitic grains in the microstructure of the SLM Ti64 part. The EDS analysis results of the SLM part are also presented in Figure 2b. The microstructure differences between the wrought and SLM Ti64 specimens can be attributed to the variations in manufacturing processes, cooling rates, and the resulting phases and crystal structures which were formed during solidification.

### 2.2. Hardness Measurement and Tensile Test

Initially, the hardness of the wrought and SLM Ti64 parts was measured using the TTS Matsuzawa HWMMT-X3 (TTS Unlimited Inc., Osaka, Japan) model Vickers hardness tester, applying a test load of 300 g for 15 sec. Five hardness measurements were taken from each sample to comply with ASTM E92-17, which is used for determining Vickers and Knoop hardness [37]. Tensile test specimens were designed for the wrought and SLM Ti64 alloys, as shown in Figure 3a, to perform tensile tests. The wrought Ti64 tensile test specimens were obtained using a wire electron discharge machine (JSEDM W-A30, Taiwan, China), while the SLM Ti64 specimens were manufactured according to the g-code generated by the slicing program.

### 2.3. Micro-Milling Experiments

The micro-milling tests were conducted on the wrought and SLM Ti64 parts using a CNC micro-milling center. The micro-milling centre has a maximum power of 2.2 kW and a top spindle speed of 60,000 rpm (Figure 4a). To determine the micro-cutting forces, a Kistler 9119AA1 model mini dynamometer was employed (Figure 4b). The dynamometer provided high precision and helped to maintain a nearly constant ambient temperature. The coordinate system of the mini dynamometer is illustrated in Figure 4b. The tests utilized coated AlTiSiN cutting tools with a diameter of 500 μm, which were supplied by the Quality Tooling System Company (Medina, OH, USA). The geometrical and cutting parameters used in the experiments are presented in Figure 5. Prior to the micro-milling experiments, the edge radius (R_e_), an important factor for the MCT, was measured to be approximately 1.7 µm.

The experimental system and micro-milling parameters which were employed in this study are presented in Table 3. In experiment series no. 1, various feed rates were utilized while maintaining a constant depth of cut and cutting speed. The objective was to determine the MCT for both workpiece materials. Test series 2 and 3 focused on investigating the effects of the depth of cut and spindle speed, respectively. The instantaneous change in force during cutting was recorded and is depicted in Figure 6a. The workpiece width is 18 mm (Figure 6b), and milling was conducted to create micro-slots across this width. The cutting distance of 18 mm remained constant for all of the experiments listed in Table 3, representing the length of a single micro-slot. To investigate the impact of the micro-milling parameters on surface roughness, surface measurements of the machined channel surfaces were conducted using a Nanovea ST400 3D optical profilometer. The ISO 25178 standard [37] was followed as the basis for all of the measurements. The scanning frequency was set at 2000 Hz with a scanning step of 1 µm. The area surface roughness of the slot was obtained by scanning the three-dimensional surface across the slot width. For each slot, at least three different measurements of a 0.6 mm length were taken (Figure 6b).

Another significant result in this experimental study was the variation in burr width between the wrought and SLM materials. The burr width formed at different cutting parameters for both materials was determined using SEM images. Microscopic analysis was performed using the FEI Quanta Feg 250 model SEM. Additionally, the SEM images were imported into the Image-J processing program to measure the average burr widths. The average burr width was determined by measuring five different burr widths from the down-milling and up-milling sides, as shown in Figure 6c.

## 3. Results and Discussions

### 3.1. Hardness Measurement and Tensile Test Results

The mechanical properties obtained from the tensile and hardness tests for the wrought and SLM materials are given in Table 4. The average hardness value for the wrought material was measured as 314 HV, while the SLM material exhibited a higher value of 368 HV. This difference can be attributed to the contrasting microstructures of the two materials. The wrought material consists of equiaxed α grains with β grains located at the grain boundaries. On the other hand, the SLM material, which is produced through additive manufacturing, exhibits an elongated β grain structure [38,39,40,41]. Consequently, this disparities in microstructural properties lead to variations in the mechanical properties of both alloys. The SLM material demonstrated a higher tensile strength (27% higher) and hardness (17% higher) compared to the wrought material. The increased strength and hardness observed in the SLM components can be attributed to their microstructure, characterized by distinctive acicular features. According to Figure 2, the wrought Ti64 specimen is typically processed through a combination of hot working and subsequent annealing. This results in a more refined microstructure with distinct α and α-β grains. The α phase refers to the pure titanium phase, while the α-β phase represents a mixture of alpha and beta phases. On the other hand, the SLM Ti64 specimen is produced using a powder bed fusion additive manufacturing technique, where a laser selectively melts and fuses successive layers of powdered titanium alloy. The rapid heating and cooling rates associated with this process often lead to the formation of a unique microstructure. In the present study, the microstructure shows acicular α martensitic grains, indicating a metastable phase which was formed due to the rapid solidification and cooling. Martensite is a distinct crystal structure that is typically characterized by a needle-like or plate-like morphology [42].

### 3.2. Cutting Force Results

Figure 7a demonstrates the variation in peak-to-valley of F_x_ and F_y_ depending on the feed per tooth at a constant cutting depth and spindle speed. It should be noted that ploughing occurs when the feed is smaller than the MCT due to the size effect in micro-milling [37,43,44]. As shown in Figure 7a, the cutting forces increase when f_z_ < 1 mm, which is consistent for both SLM and wrought Ti64 materials. This suggests that the MCT for both materials is 1 µm/tooth. Previous studies in the literature have also indicated that cutting forces exhibit irregular behavior at feed rates lower than the MCT [30]. The region smaller than f_z_ < 1 µm/tooth is defined as the ploughing zone, while the range of 1 µm/tooth < f_z_ < 5 µm/tooth is defined as the transition region. In the transition region, the cutting forces experience fluctuations. However, the magnitude of change is not as significant as in the ploughing zone.

Interestingly, when comparing the shear forces between the two materials, an unexpected trend is observed considering the hardness of the test specimens. It was anticipated that SLM, which has a higher strength and stiffness, would result in higher cutting forces. However, the opposite was observed, where SLM exhibited lower cutting forces. This result aligns with other studies on the micro-milling of Ti alloys, where cutting forces are generally smaller for hard material [30,31]. Another significant observation from Figure 7a is that the production technique has no effect on the MCT. Both the wrought and SLM materials show a similar trend in cutting forces with a varying feed, and a consistent MCT value of 1 µm is obtained.

Figure 7b also exhibits changes in the F_x_ and F_y_ forces with a depth of cut for f_z_ = 4 μm/tooth and *n* = 35,000 rpm. As seen in Figure 7b, the cutting forces increase linearly with an increase in the depth of the cut. According to cutting mechanics, this increase in cutting forces can be explained by the increased chip cross-section. Furthermore, higher force amplitudes were obtained for the wrought Ti64 sample compared to the SLM part during milling. This confirms that the forces required for processing the wrought product must be greater than those for the SLM. At depths of cut greater than 100 µm, an increasing trend in cutting forces is observed more prominently in both samples. As seen in Figure 7b, the F_x_ and F_y_ for wrought Ti64 are 4.8 N and 3.7 N, respectively, at a depth of cut of 25 μm. For the same depth of cut, the F_x_ and F_y_ values in the SLM material are 3.7 N and 3 N, respectively. However, in wrought Ti64 material, for a_p_ = 200 μm, the F_x_ and F_y_ forces are 12.6 N and 10.2 N, respectively, while for SLM Ti64 they are 12.1 N and 8.8 N. Alatrushi [45] compared the micro-machinability of Ti64 and Ti 5553 alloys and stated that the cutting forces increased with an increasing depth of cut. It was also explained that a higher cutting force is needed to cut material with lower hardness in the ploughing stage [46]. In addition, Hojati et al. [31] stated that more energy is required to cut wrought Ti64 compared to Ti64 produced with EBM.

Figure 7c shows the influence of the spindle speed on the cutting forces for f_z_ = 1 µm/tooth and a_p_ = 100 µm. The cutting forces obtained for both Ti64 alloys exhibit a different trend with increasing spindle speed. The cutting forces experienced by the wrought alloy at *n* = 5000 rpm are approximately twice as high as those for the SLM material. The reason why SLM material exhibits lower shear force despite its higher strength and hardness may stem from the total force involved in cutting, including the cutting energy consumption for pure cutting, friction, and plastic deformation [31]. The wrought material has a higher toughness and ductility due to its larger grain structure, leading to increased deformation during chip formation and a higher energy consumption for plastic deformation. As can be seen in Figure 7c, the most suitable spindle speed for both test samples is 10,000 rpm. Therefore, an optimum spindle speed value of 10,000 rpm can be recommended.

### 3.3. Average Areal Surface Roughness Values

Figure 8 shows the average surface roughness (S_a_) at different feed rates, a constant depth of 100 μm, and a spindle speed of 32,000 rpm. The S_a_ values show a significant increase for f_z_ < 1 μm/tooth. In addition, the minimum S_a_ values were obtained at 2 μm/tooth for both workpiece materials. In Figure 8, a detailed view of the S_a_ changes for f_z_ < 6 μm/tooth is provided as an inset graph. The f_z_ ≤ 1 μm/tooth range represents the ploughing-dominated zone, while 1 ≤ f_z_ ≤ 2 μm/tooth represents the transition zone. The amplitudes of the error bars (the difference between the lowest and highest values) increase in the ploughing-dominated zone, indicating a highly unstable and irregular surface roughness. For f_z_ > 2 μm/tooth, the shear mechanism becomes dominant, similar to what is observed in conventional milling. The most interesting result that can be observed in Figure 8 is that the S_a_ values obtained for the SLM material in the ploughing-dominated zone are higher. However, for f_z_ > 2 μm/tooth (shear zone), the S_a_ values obtained for the SLM material are lower. In other words, in the ploughing zone, the material with the higher hardness exhibits a lower surface quality. Conversely, in the shear zone, the material with the higher hardness has a better surface quality. Generally, conventional milling results in a better surface finish on workpieces with a higher hardness [47]. This is because in more ductile workpieces, the formation of a built-up edge (BUE) is more common, which negatively affects the surface roughness. The higher S_a_ observed for the SLM material in the ploughing region may be attributed to its grain structure. The rapid solidification and cooling rates associated with SLM can result in a non-uniform distribution of grain sizes and orientations within the material [48]. During the micromachining process, especially in the ploughing zone, the cutting tool interacts with the material, causing plastic deformation and material removal. The presence of fine equiaxed grains in the SLM material can lead to increased grain boundary effects and grain orientation variations. These variations can contribute to localized variations in material hardness and mechanical properties, which can result in a higher surface roughness (S_a_) in the ploughing-dominated zone. In a study by Airao et al. [49], discontinuous chip formation was commonly observed in SLM material under micro-turning conditions. From this perspective, the presence of discontinuous micro-chips provides an opportunity for the ploughing effect to smear the machined surface more easily, particularly when the feed is smaller than f_z_ < 1 μm/tooth. This leads to an increase in S_a_ and a decrease in surface quality.

In Figure 9, 3D surface topographies of the machined surfaces of SLM and wrought materials are given for three different feed rates. Figure 9 displays the maximum surface roughness measured in the ploughing region, along with the corresponding images at the maximum feed value. Furthermore, the result for f_z_ = 2 µm/tooth, where the minimum S_a_ values are achieved for both materials, is also provided. The ploughing effects are clearly visible in the surface images which were obtained for f_z_ = 0.05 µm/tooth in both materials. These cutting marks are irregular due to the tool’s sharp edge attempting to cut the chip in the next revolution when it was unable to cut it in the previous revolution. As the feed value increases, both the irregular marks on the machined surface decrease, and the difference between the peak and trough diminishes. This trend is clearly depicted in the scale provided next to the topography. For both materials, the marks left by the cutting edge are more pronounced at f_z_ = 10 µm/tooth. It can be observed that the machining marks are more distinct for f_z_ ≥ 2 µm/tooth in the material produced by the SLM method.

Figure 10 illustrates the effects of the cutting depth and spindle speed on surface roughness. At low depths of cut, the difference between SLM and wrought material is at the maximum and a better surface is obtained in SLM material (Figure 10a). As the depth of cut increases, the difference in S_a_ values decreases, and for a_p_ = 200 µm, the S_a_ values obtained for both materials become very close to each other. In addition, the amplitudes of the error bars increase for a_p_ ≤ 50 µm. In a study by Aslantas et al. [50], it was noted that the S_a_ values remained constant for the wrought Ti64 alloy at a depth of cut of up to 200 µm. In this study, it can be inferred that the S_a_ values do not change significantly or remain relatively constant as the depth of cut increases. In Figure 10b, the S_a_ variations with the increasing spindle speed are given. S_a_ values tend to decrease with an increasing spindle speed for both materials. A decrease in the spindle speed corresponds to a decrease in the cutting speed. It is well-known that the likelihood of BUE formation increases at medium and low cutting speeds [51]. In this study, it is thought that the factor which caused the increase in S_a_ values at speeds lower than *n* < 10,000 rpm was BUE formation. Another result which was obtained is that the S_a_ values obtained in the SLM and wrought materials for *n* ≥ 10,000 rpm are very close to each other.

### 3.4. Burr Formation

Burr formation on workpieces after micro milling is a complex and troublesome problem. The presence of burrs requires additional operations such as deburring, which pose a risk of damaging the workpiece [44,52,53,54]. Optimizing the cutting parameters during micro-milling can help to minimize burr formation [55]. In addition, in micro-milling, the formation and size of burrs can vary depending on the tool’s cutting direction [56]. In other words, the burr widths formed during up-milling and down-milling may differ. In this study, the influence of the micro-milling parameters on the top burr width was investigated for both wrought and SLM Ti64 materials.

### 3.5. Effect of Micro-Milling Parameters on Burr Widths

In Figure 11, the variation in burr width on the up- and down-milling sides is shown, depending on the feed rate per tooth. The burr width on the down-milling side is 68% larger than that on the up-milling side. Down-milling refers to the direction in which the cutting-edge leaves the workpiece. Since the chip thickness decreases with the tool’s rotational motion, the cutting-edge radius acts as a negative rake angle. Hence, the micro-milling tool shows ploughing rather than material removal. This situation also leads an increment in burr width on the down-milling side [57,58,59]. Another result shown in Figure 11 is that the maximum burr width occurs at f_z_ = 0.5 µm/tooth in both milling directions. This aligns with the results shown in Figure 7a, in which the maximum shear force values were measured for f_z_ = 0.5 µm/tooth. The results shown in Figure 7a and Figure 11 confirm each other, indicating that the maximum ploughing occurs at f_z_ = 0.5 µm/tooth. Furthermore, as the feed rate increases, the burr width, particularly on the down-milling side, shows a significant decreasing trend. For feed rates higher than f_z_ = 5 µm/tooth, the burr widths on both the down- and up-milling sides remain constant.

In Figure 12, SEM images of burr widths at different feed rates per tooth for wrought and SLM materials are presented. The ratio of feed per tooth to tool edge radius is also indicated in Figure 12. It can be observed that the maximum top burr width for both materials occurs at f_z_ = 0.5 µm/tooth. As the feed rate increases, the burr width decreases. Specifically, in the case of f_z_ < 0.5 µm/tooth, the burr width in SLM Ti64 is smaller compared to the wrought material. This observation aligns with the results in Figure 11. However, for f_z_ = 10 µm/tooth, the burr in SLM Ti64 is negligible.

Figure 13 presents the change in the burr width depending on the cutting depths on the up- and down-milling side. At all depths of cut and on both milling sides, the burr width obtained in the SLM material is lower than in the wrought material. Even in up-milling, the burr width in the SLM material appears to be independent of the cutting depth. However, a similar trend was not observed in the case of up-milling. In addition, the burr widths obtained for the wrought material exhibit significant variation, as indicated by the error bars. This variation can be attributed to the material’s more ductile behavior. Since plastic deformation is less common in the machining of SLM material (due to its hardness being higher) than in wrought material, narrower and finer burrs can be expected when machining SLM material [31]. These findings are in agreement with the outcomes of micro-ball end milling Ti64 presented by Chen et al. [60]. It is known that, as the depth of cut rises, the contact area between the workpiece and the tool goes up as well, the extrusion and ploughing process intensifies, and the cutting forces increase. Therefore, more material deformation is induced and bigger top burrs are shaped through the cutting edge [61]. SEM images of burr widths at different cutting depths for both materials are given in Figure 14. It can be seen that the burr width in the SLM material is significantly lower. According to the results shown in Figure 13 and Figure 14, it is suggested to consider the depth of cut as 100 µm for the maximum material removal rate in SLM material.

The variation in the burr widths formed on the up- and down-milling sides with an increasing spindle speed is given in Figure 15. Increasing the spindle speed also causes the burr widths to increase. Similar to the results obtained in Figure 11, it is possible to say that the burr widths obtained in down-milling are higher. As the spindle speed increases, the friction and extrusion time on the up-milling side is shortened, resulting in a smaller burr size. However, on the down-milling side, the chip becomes harder to break and stays above the slot due to thermal softening influence during micro-milling. This causes the burr width to be higher. However, contrary to the results obtained in Figure 13, the difference between SLM and wrought material is minimal. The results obtained in this study are consistent with those obtained by Özel et al. [62]. Aslantas et al. [63] also state that the spindle speed increases the burr width in the micro-milling of Ti64 alloy. However, Wu et al. [64] emphasize that the burr width decreases with an increasing spindle speed for pure copper material. It is thought that this difference is due to the work material. Figure 16 shows SEM images of the top burrs for different spindle speed values. In the wrought material, the burrs take a more needle-like shape with an increasing spindle speed. Conversely, in SLM material, the burrs are more irregular at higher spindle speed values and, at 50,000 rpm, the burrs become wavy and more continuous.

## 4. Conclusions

This study investigated the micro machinability of Ti-6Al-4V (Ti64) parts produced via selective laser melting (SLM) compared with wrought Ti64. The study focused on the effect of micro-milling parameters on cutting forces, surface roughness, and burr widths.

⮚The microstructures of Ti-6Al-4V (Ti64) parts produced via selective laser melting (SLM) differ from those produced via traditional processes. SLM parts exhibit acicular α martensitic grains, leading to a higher hardness and tensile strength compared to wrought Ti64.⮚The micro machinability of SLM Ti64 parts was compared with wrought Ti64 using micro-milling parameters. The minimum chip thickness (MCT) for both SLM and wrought Ti64 was found to be 1 μm/tooth.⮚The cutting forces in micro-milling increased for feed rates smaller than the MCT, indicating a ploughing zone. The cutting forces in the transition zone (1 μm/tooth < f_z_ < 5 μm/tooth) increased and decreased, but the change was not as significant as in the ploughing zone.⮚The production technique (SLM or wrought) did not affect the MCT. Both SLM and wrought Ti64 exhibited similar trends in cutting forces with varying feed rates.⮚The cutting forces increased linearly with the depth of cut. Wrought Ti64 required higher cutting forces compared to SLM Ti64, indicating that more force is needed to machine the wrought material.⮚The influence of the spindle speed on the cutting forces varied for SLM and wrought Ti64. SLM material exhibited lower shear forces despite its higher strength and hardness compared to wrought material.⮚The surface roughness (S_a_) increased for feed rates smaller than 1 μm/tooth, exhibiting a ploughing-dominated zone. For feed rates larger than 2 μm/tooth, SLM Ti64 exhibited lower S_a_ values compared to wrought Ti64, indicating a better surface quality.⮚The grain structure of the SLM material and the presence of fine equiaxed grains contributed to higher surface roughness in the ploughing-dominated zone. Discontinuous chip formation in SLM material also contributed to increased surface roughness.⮚The variation in surface roughness with the cutting depth and spindle speed showed that the Sa values remained relatively constant with an increasing depth of cut. Lower spindle speeds resulted in higher Sa values due to the increased probability of built-up edge formation.⮚Burr formation during micro-milling was a complex issue. Optimizing cutting parameters can help to minimize burr formation. The influence of micro-milling parameters on burr widths in SLM and wrought Ti64 was studied.

These conclusions highlight the differences in mechanical properties, cutting forces, surface roughness, and burr formation between SLM and wrought Ti-6Al-4V materials during micro-milling. These findings provide valuable insights for the manufacturing of functional parts with complex geometries using SLM and subsequent micro-milling processes.

## Figures and Tables

**Figure 1 micromachines-14-01160-f001:**
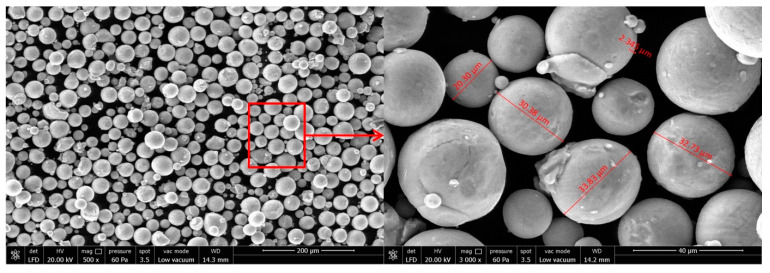
SEM view of the Ti-6Al-4V powders.

**Figure 2 micromachines-14-01160-f002:**
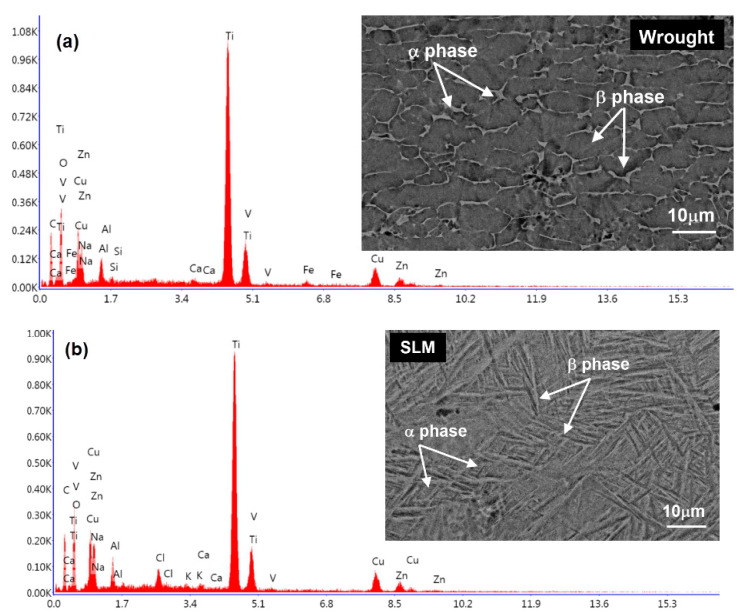
Microstructure and EDS analysis of (**a**) wrought Ti-6Al-4V and (**b**) SLM Ti-6Al-4V.

**Figure 3 micromachines-14-01160-f003:**
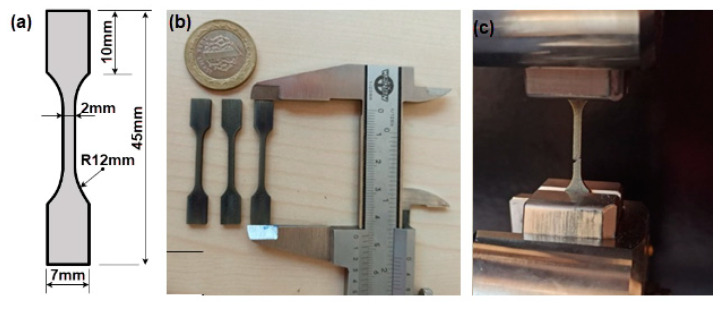
Tensile test specimens and tensile test procedure: (**a**) dimensions of the tensile test specimens (mm), (**b**) manufactured Ti-6Al-4V tensile test specimens, and (**c**) tensile testing of the Ti-6Al-4V tensile test specimens.

**Figure 4 micromachines-14-01160-f004:**
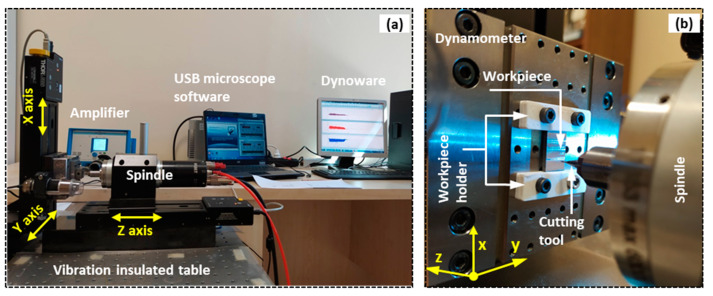
High precision micro machining system, (**a**) general view of the micro-milling test equipment and (**b**) exhibition of the micro tool, workpiece, and dynamometer with details.

**Figure 5 micromachines-14-01160-f005:**
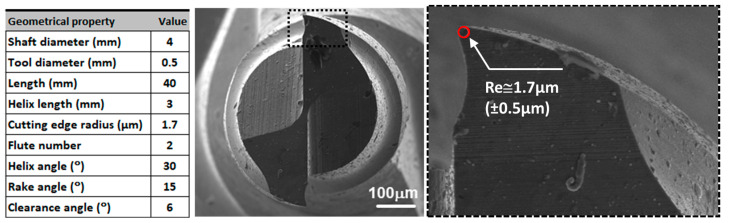
Geometric properties and SEM image of the cutting tool used in micro-milling tests.

**Figure 6 micromachines-14-01160-f006:**
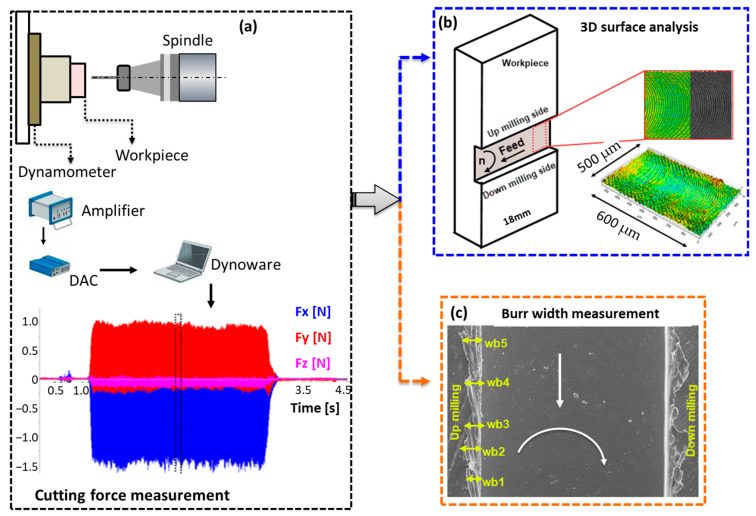
Experimental methodology used in micro milling tests.

**Figure 7 micromachines-14-01160-f007:**
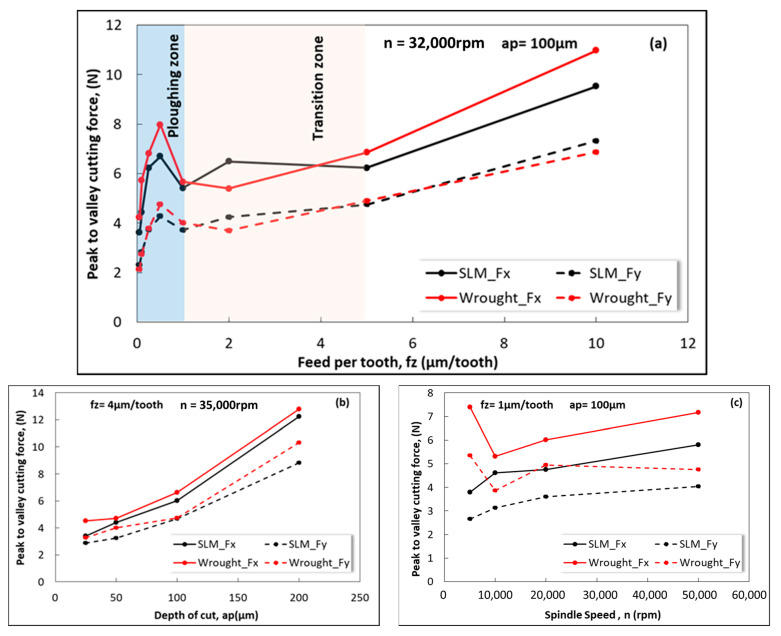
Variation in peak-to-valley cutting forces during micro-milling of SLM and wrought Ti-6Al-4V parts depending on (**a**) feed per tooth, (**b**) depth of cut, and (**c**) spindle speed.

**Figure 8 micromachines-14-01160-f008:**
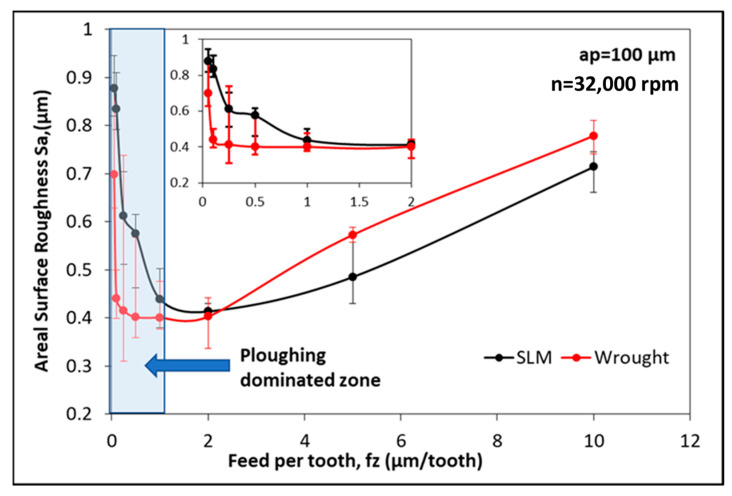
Variation in surface roughness with feed per tooth for SLM and wrought material.

**Figure 9 micromachines-14-01160-f009:**
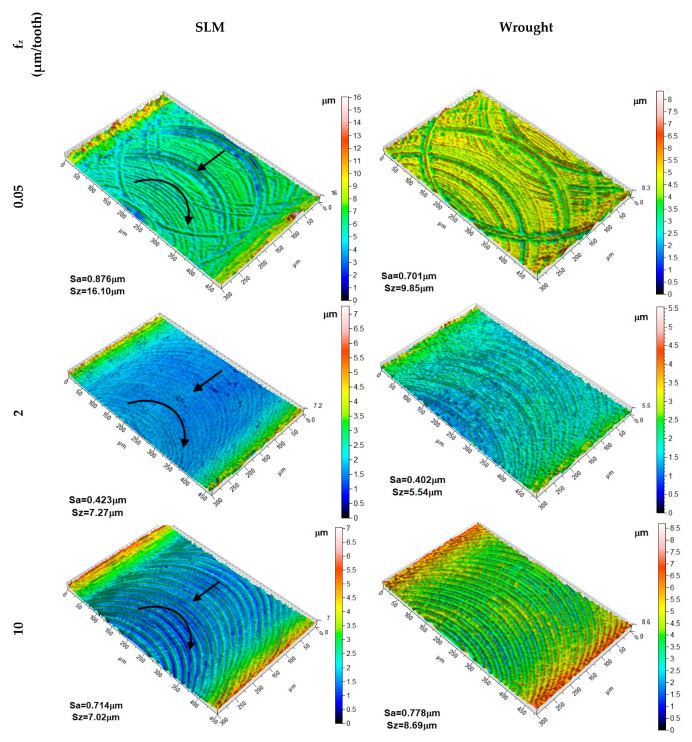
3D surface topography of the machined slot for SLM and wrought material (a_p_ = 100 µm, *n* = 32,000 rpm).

**Figure 10 micromachines-14-01160-f010:**
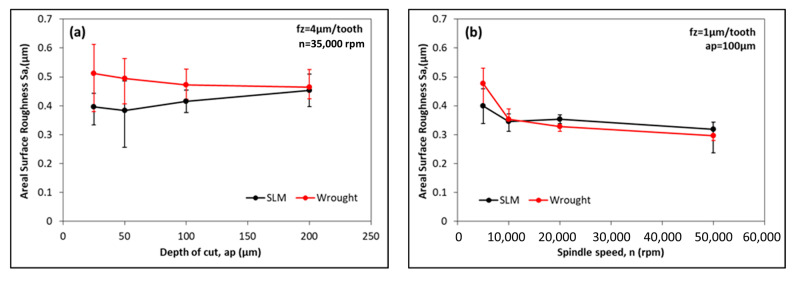
Variation in S_a_ values for SLM and wrought material with (**a**) depth of cut and (**b**) spindle speed.

**Figure 11 micromachines-14-01160-f011:**
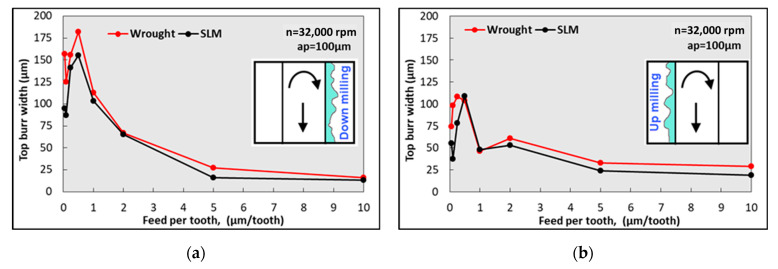
Top burr width variation in down- and up-milling for both materials; (**a**) down-milling and (**b**) up-milling.

**Figure 12 micromachines-14-01160-f012:**
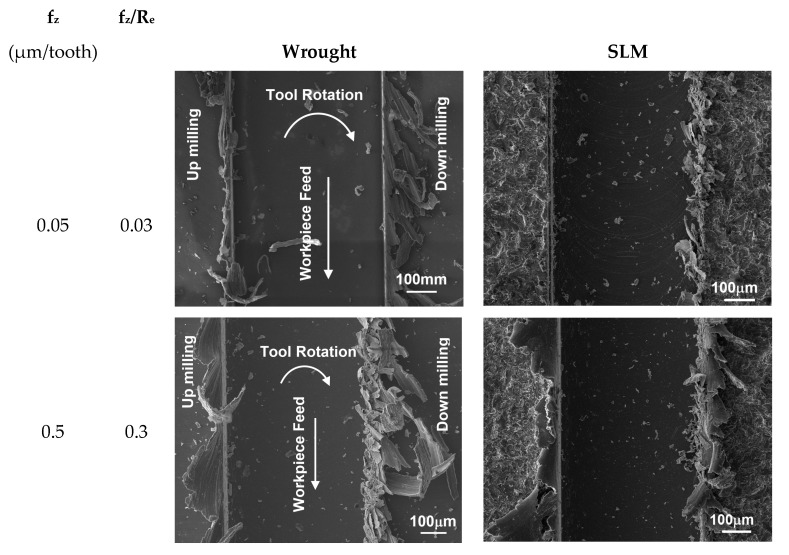
SEM images of micro slots for different feed rates.

**Figure 13 micromachines-14-01160-f013:**
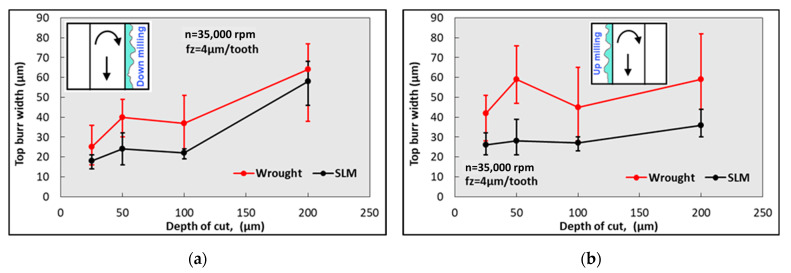
Variation in burr width values depending on depth of cut; (**a**) down-milling and (**b**) up-milling.

**Figure 14 micromachines-14-01160-f014:**
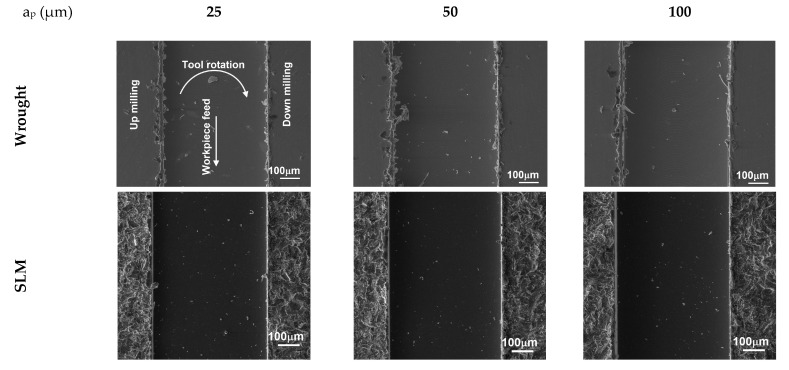
SEM images of micro slots for different depths of cut.

**Figure 15 micromachines-14-01160-f015:**
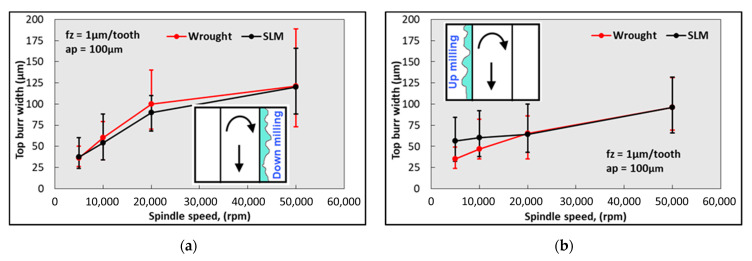
Variation in burr width values depending on spindle speeds; (**a**) down-milling and (**b**) up-milling.

**Figure 16 micromachines-14-01160-f016:**
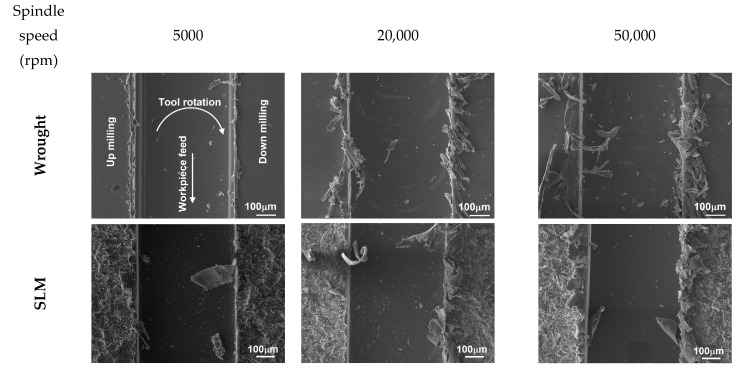
SEM images of micro slots for different spindle speeds.

**Table 1 micromachines-14-01160-t001:** Chemical composition and physical features of the Ti6Al4V powders.

Chemical Composition	Ti	Al (%)	V (%)	O	N	C	H	Fe
	Main Element	5.5–6.75	3.5–4.5	<2000 ppm	<500 ppm	<800 ppm	<150 ppm	<3000 ppm
Physical properties	Density (g/cm^3^)				Particle size (µm)			
	4.42				30–50			

**Table 2 micromachines-14-01160-t002:** SLM process parameters used during the manufacturing of Ti-6Al-4V workpiece.

Layer Thickness (µm)	Laser Power (W)	Scanning Speed (mm/s)	Hatch Distance (mm)	Energy Density (J/mm^2^)
30	170	1250	0.1	1.36

**Table 3 micromachines-14-01160-t003:** Cutting parameters used in experiments.

Experiment Number	Spindle Speed, *n* (rpm)	Cutting Speed (m/min)	Feed per Tooth, f_z_ (µm/Tooth)	Depth of Cut, a_p_ (µm)	Cutting Length (mm)	Aim of the Experiment
1	32,000	50.26	0.05, 0.1, 0.25, 0.5 1, 2, 5, 10	100	18	Determination of the minimum chip thickness. Effect of the feed per tooth on surface roughness and burr formation
2	35,000	54.9	4	25, 50 100, 200	18	Effect of depth of cut on cutting force, surface roughness and burr formation
3	5000 10,000 20,000 50,000	7.85 15.7 31.4 78.5	1	100	18	Effect of spindle speed on cutting force, surface roughness and burr formation

**Table 4 micromachines-14-01160-t004:** Mechanical properties of SLM and wrought Ti-6Al-4V.

	SLM Ti-6Al-4V	Wrought Ti-6Al-4V
Hardness (HV)	368 ± 10.2	314 ± 8.6
Tensile Strength (MPa)	1086 ± 26.1	850 ± 18.4
Elongation at break (mm)	2.770	2.118

## Data Availability

Not applicable.

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
