# Peer review of "Comparative Analysis of Minimum Chip Thickness, Surface Quality and Burr Formation in Micro-Milling of Wrought and Selective Laser Melted Ti64"

_micromachines, 2023, doi:10.3390/mi14061160_

Round 1
Reviewer 1 Report
Paper 'Comparative analysis of minimum chip thickness, surface quality and burr formation in micro-milling of wrought and selective laser melted Ti64' is aiming to study the influence of micro milling parameters on resulting cutting forces, surface roughness and burr width for the SLM-ed and wrought Ti64. Despite good writing, there were some questions that should have been added prior to acceptance.
1 In Figure2, the microstructure of wrought and SLM Ti64 is obviously different. The authors should explain the reason for the difference.
2 In line 179, ‘Therefore, this difference in microstructural properties causes differences…’, whether the elongated β grain structure is the main reason for the enhancement of mechanical properties?
3 In line 251-252, ‘The reason for the higher Sa for SLM material in the ploughing region may be the grain structure’, why the grain structure of SLM material leading to the higher Sa?
Reviewer 2 Report
This manuscript presents a study on the micro milling process for selective laser melted (SLM) Ti-6Al-4V alloys. The study investigates various milling process parameters and their effects on cutting force, surface roughness, and burr formation. While the study is well designed and executed, there are some areas that need improvement for publication in 'micromachines'. The following suggestions aim to address these areas:
Implication of Results in the Conclusions Section: It is recommended to include a discussion in the conclusions section regarding the implications of the results. Specifically, the authors should emphasize the significance of micro milling SLM parts in comparison to wrought Ti-6Al-4V milling performance. This will provide a clear connection between the study's findings and their potential impact on the development of micro milling devices, as indicated in the introduction section.
Description of Tool Path and Axis Configuration: The authors should provide a detailed description of the tool path used during the micro milling process, along with the configuration of the axis. This information will enhance the clarity of the study.
Illustration of Cutting Force Measurement with Timescale Results: To improve the understanding of cutting force measurements, it would be beneficial to provide an example that illustrates how the cutting force is measured, preferably with a timescale result. This will help readers visualize the process of measuring cutting forces.
Consistent Size and Alignment of Graphs in Figure 2: Ensure that all graphs in Figure 2 have the same size and alignment. This will provide a visually cohesive presentation of the data, making it easier for readers to compare and interpret the results.
Same Color Bar Range for Comparing Results in Figure 9: To facilitate better comparison of results, it is recommended to use the same color bar range for all data points in Figure 9. This will ensure that the color scale remains consistent across the graph, allowing readers to accurately compare different regions.
Use Straight Lines for Plots in Figures 13 and 15: For clarity and visual simplicity, it is suggested to use straight lines for the plots in Figures 13 and 15. Straight lines can provide a more precise representation of the data, enabling readers to discern trends and patterns more easily.
Consistent Font Size and Type in Figures and Captions: It is important to maintain consistency in font size and type throughout the manuscript, particularly in figures and their captions. Using the same font size and type will enhance readability and make the manuscript more professional in appearance.
Correct Usage of Subscripts: Carefully review the manuscript to identify and correct any instances where subscripts, such as H2O, Fx, Fy, Sa, and fz, are not properly formatted. Ensure that subscripts are appropriately displayed to maintain accuracy and clarity.
Improving English Writing with Professional Support: A professional proofreader or editor can help enhance the clarity, flow, and grammar of the text, resulting in a more polished final manuscript.
